# Targeting the Exon2 splice cis-element in PD-1 and its effects on lymphocyte function

Yuto Tan[1], Naoko Kumagai-Takei(iD)[2]*, Yurika Shimizu[2], Akira Yamasaki[2], Mari Hara-Yamamoto[3], Shigeru Mitani[1], Tatsuo Ito[2]

1 Department of Bone and Joint Surgery, Kawasaki Medical School, Kurashiki, Japan, 2 Department of Hygiene, Kawasaki Medical School, Kurashiki, Japan, 3 Okayama University Medical School, Okayama, Japan

* kumagai@med.kawasaki-m.ac.jp

## Abstract

T-cell therapies have proven to be a promising treatment option for cancer patients in recent years, especially in the case of chimeric antigen receptor (CAR)-T cell therapy. However, the therapy is associated with insufficient activation of T cells or poor persistence in the patient's body, which leads to incomplete elimination of cancer cells, recurrence, and genotoxicity. By extracting the splice element of PD-1 pre-mRNA using biology based on CRISPR/dCas13 in this study, our ultimate goal is to overcome the above-mentioned challenges in the future. PD-1 plays an important role in controlling T cell responses and is expressed at the cell surface of T cells following activation. The receptor PD-1 interferes with T cell receptor (TCR) signaling following interaction with PD-L1. The outcome of stimulation via PD-1 leads to decreases in cytokine secretion and cell proliferation. We extracted the RNA region of PD-1 pre-mRNA using CD8+T cell lines and examined the effect of targeting the Exon2 splice cis-element on the production of cytokines in the present study. In particular, the production of IFN-γ, TNF-α, GM-CSF was lower in RNA-targeted cells than in non-targeted cells, but the cytokine secretion capacity and cell proliferation were maintained in RNA-targeted cells. These results suggested that the use of the RNA editing technology, CRISPR/dCas13 strategy offers a novel approach to mitigate genotoxicity in lymphocytes with cytokine production and cell proliferation.

## Introduction

Programmed Cell Death 1 (PD-1) is a type I transmembrane protein preferentially expressed on immune cells including T, B, and NK cells. Its ligand, Programmed Cell Death 1 Ligand 1 (PD-L1), is expressed in various cell types including cancer cells. PD-L1 is also expressed on antigen-presenting cells and is a member of the B7 family. B7 family members regulate immune responses by delivering co-stimulatory or co-inhibitory signals [1].

**Data availability statement:** All relevant data are within the manuscript and its Supporting Information file.

**Funding:** N.K.-T. In full Naoko Kumagai-Takei This work was supported by JSPS KAKENHI (JP22K10497) and Research Project Grants (R03B039, R04B042, R05B043, R06B029) from Kawasaki Medical School.

**Competing interests:** The authors have declared that no competing interests exist.

Upon interaction with PD-L1, the receptor PD-1 potently interferes with T cell receptor (TCR) signaling through intracellular molecular mechanisms. The protein structure of PD-1 consists of an extracellular immunoglobulin variable region (IgV)-like domain, a transmembrane region, and a cytoplasmic domain that contains immunoreceptor tyrosine-based inhibitory motifs (ITIMs) and an immunoreceptor tyrosine-based switch motif (ITSM) [2]. Interference with PD-1 signaling by immune checkpoint inhibitors enhances T cell function by enhancing signal transduction from the TCR signalosome [3].

PD-1 plays an important role in controlling T cell responses. PD-1 expression is induced at the cell surface of T cells following activation. Constitutive PD-1 expression by tumor-specific T cells is known to be associated with the expression of additional inhibitory receptors, leading to impaired T cell function and tumor escape, upon ligation to its ligand PD-L1 expressed by tumor cells or immune infiltrating cells within tumor microenvironments [4]. In the antigen-specific T cell response, Zap70 is phosphorylated following binding to the T cell receptor. However, this phosphorylation is counteracted by the interaction of PD-1 with PD-L1. The outcome of stimulation via PD-1 leads to decreases in cytokine secretion and cell proliferation [5,6]. Our research hypothesis holds that depletion of lymphocytes through this PD-1-mediated mechanism may lead to decreased efficacy in the treatment for cancer patients.

In recent years, T-cell therapies have proven to be a promising treatment option for cancer patients, especially in the case of chimeric antigen receptor (CAR)-T cell therapy. However, current challenges include insufficient activation of T cells or poor persistence in the patient's body, which can lead to incomplete elimination of cancer cells and recurrence. As an alternative method, the permanent removal of PD-1 using genome editing (e.g., via CRISPR/Cas9) is considered, although knockout of PD-1 in therapeutic T cells carries high risks such as the development of tumor cells associated with genotoxicity [7]. In an effort to avoid these risks, methods involving temporary inhibition of PD-1 expression have been investigated. In this study, we extracted an RNA region to target PD-1 pre-mRNA and temporarily inhibit PD-1 expression using our CRISPR/dCas13 system. We also examined the effect of targeting the Exon2 splice cis-element on lymphocyte function, focusing particularly on cytokine production.

## Materials and methods

### Cell culture of the human CD8⁺ T cell line

The human CD8$^+$ T cell line EBT-8 was a gift from Prof. H. Asada [8] and was maintained in GIT medium (FUJIFILM Wako Pure Chemical Corporation, Osaka, Japan) supplemented with 80 U/ml recombinant human IL-2 (KYOWA Pharmaceutical Industry Co., Ltd., Osaka, Japan), 100 µg/ml streptomycin, and 100 U/ml penicillin (Meiji Seika Pharma Co., Ltd., Tokyo, Japan). EBT-8 was established simply by continuously culturing mononuclear cells obtained from a patient with leukemia, which means that EBT-8 was not established as a specific T cell clone for a specific

antigen. Cells were placed in $25\,cm^2$ tissue culture flasks in portrait style in a volume of $10\,ml$, and incubated at 37°C in a humidified atmosphere of 5% carbon dioxide in air.

### Guide RNA (gRNA) design

gRNA design was performed as described by Bandaru et al [9]. Briefly, a total of 18 different gRNAs were designed with SnapGene Viewer 4.1.9. Prior to oligosynthesis, BbsI restriction sequence was added at the 5′end of the designed gRNAs. All the gRNA oligonucleotides were procured from Integrated DNA Technologies, Inc. (IDT) (Coralville, IA). Details of the gRNAs designed for the study are provided in supplementary form (Supporting Information-1 (gRNAs for PD-1)).

The gRNA oligonucleotides were subcloned into the backbone vector (pLKO5.U6.crRNA.tRFP) expressing RFP following restriction enzyme treatment with BsmbI.

### Transient transfection assays

dCas13 plasmid vector and 18 different gRNA expression vectors with RFP were prepared. Transformation was performed using *E. coli* strain DH5-alpha competent cells. Bacterial cells were harvested and pelleted by centrifugation after preparation of large overnight culture. Plasmid DNA was extracted from this bacterial culture using the NucleoBond Xtra Maxi Kit (MACHEREY-NAGEL GmbH & Co. KG, Düren, Germany) according to the manufacturer's instructions. To deliver vectors into EBT-8 cells, electroporation was performed using NEPA21 Type II (Nepa Gene Co., Ltd. Chiba, Japan).

### Cell isolation and DNA extraction

Following the staining of EBT-8 cells with anti-human PD-1-APC antibody (BioLegend, Inc., San Diego, CA), PD-1 positive and negative RFP⁺ cells were isolated by flow cytometry with FACS Aria III (Becton Dickinson, Franklin Lakes, NJ). PD-1 positive and negative RFP⁺ cell DNA was isolated using Genomic DNA from Tissue (Macherey-Nagel GmbH & Co. KG, Düren, Germany).

### Detection of gRNAs

gRNA regions were amplified by PCR using DNA extracted from cells as templates. The sequences of the gRNAs were identified by next-generation sequencing (Azenta Life Sciences, Burlington, MA). A new screening system was developed and used to identify the gRNAs detected in each RFP⁺ cell population (PD-1 positive and negative cells) and to calculate the number of gRNA reads.

### Assay for expression levels of cell surface molecules

EBT-8 cells were stained with PD-1-PE antibody (BioLegend, Inc.) to examine the expression levels of PD-1. The percentage of cells positive for mean fluorescence intensity was analyzed using a FACSCalibur™ (Becton Dickinson) flow cytometer.

### Multiplex cytokine/chemokine analysis

One day following electroporation with or without dCas13 plasmid vector and one type of gRNA vector, cells were stimulated with bead-bound antibodies. Monoclonal antibody CD3 (Beckman Coulter, Inc., Brea, CA) was incubated with anti-mouse IgG-coated beads (Spherotech, Inc., Lake Forest, IL) at room temperature for 30 min. After washing the beads with PBS, $2.0 \times 10^4$ cells were incubated with $2.0 \times 10^4$ beads in GIT medium supplemented with 80 U/ml recombinant human IL-2, 100 µg/ml streptomycin and 100 U/ml penicillin in 96-well round-bottomed plates. After the plates were incubated at 37°C for 48 h in a humidified atmosphere of 5% CO2, culture supernatants were collected and assayed for cytokines and chemokines. The assay was performed using a human cytokine/chemokine magnetic bead panel kit (EMD Millipore Corporation, Billerica, MA) according to the manufacturer's instructions. Data was collected using a Luminex-200 Instrument System (Luminex, Austin, TX).

## Statistical analysis

Significant differences were determined using Student's t-test and are indicated by asterisks (*P<0.05, **P<0.01).

## Results

### Expression levels of PD-1 in a human CD8⁺ T cell line, EBT-8 cells, and gRNA design

The EBT-8 cell line was established from a large granular lymphocyte leukemia of T cell origin and shows surface expression of CD2, CD3, CD8, HLA-DR, and T cell receptor alpha/beta, which are characteristic of cytotoxic T lymphocytes [8]. In this study, we examined the expression of PD-1, which is not yet known in cell lines. Flow cytometry revealed that PD-1 is expressed on the surface of EBT-8 cells (Fig 1A). PD-1 acts by down-regulating the immune system [10], and Exon2 contains the binding domain for PD-L1/PD-L2, which are PD-1 ligands. We hypothesized that disrupting the interaction of the Exon2 splice cis-trans element on PD-1 pre-mRNA might lead to lymphocytes exerting their inherent functions. In an effort to regulate RNA splicing using our CRISPR/dCas13 system, it was important to identify an RNA sequence that can guide dCas13 to a specific region of the pre-mRNA. Therefore, in this study, we designed 18 different guide RNAs that target Exon2 splice cis-elements in pre-mRNA of PD-1 (Fig 1B). The guide RNAs were designed based on the following three considerations. Some of the gRNAs needed for CRISPR/dCas13 were designed to bind to GU and AG sequences. gRNAs were also designed to target splicing factor binding sites on the PD-1 pre-mRNA. It is better to avoid designing gRNAs that recognize RNA sequences where the protospacer flanking site (PFS) contains a G at the 3'-end [9].

A

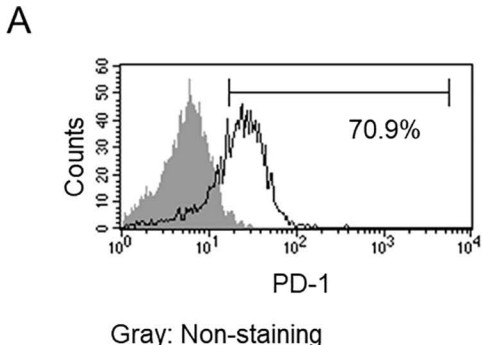

Gray: Non-staining

B

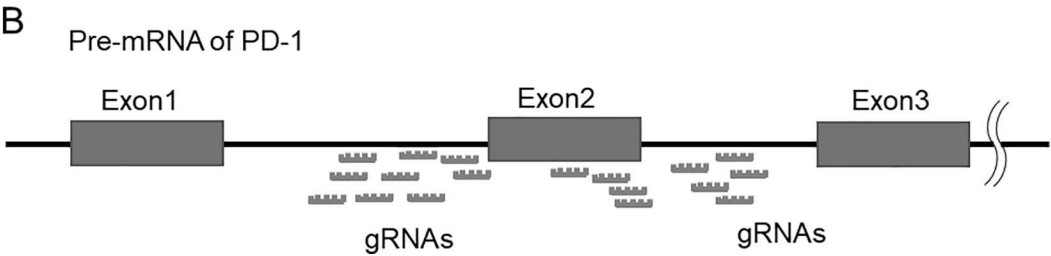

**Fig 1. gRNAs targeting the Exon2 splice cis-element in pre-mRNA of PD-1. A)** Representative histograms of cell surface PD-1 in EBT-8 cells. The number in each histogram shows the percentage of positive cells. A nonstained control (gray) is shown in the panel. B) Schematic representation of gRNAs targeting known spliceosome binding regions in pre-mRNA of PD-1.

## Screening of gRNAs using the CRISPR/dCas13 system

If the gRNA sequence that regulates the RNA splicing can be identified using dCas13 and the 18 gRNAs shown in Fig 1B above, dCas13 with the identified gRNA will bind to that region and inhibit the splicing of Exon 2. In other words, Exon 2 will not be translated and the extracellular domain of the PD-1 receptor will be absent. To conduct screening of gRNAs using the CRISPR/dCas13 system, gRNA-transfected cells in PD-1-negative cells or PD-1-positive cells were sorted (Fig 2A). The distribution of the different gRNAs was highly heterogeneous (Fig 2B). For instance, in PD-1 negative cells,

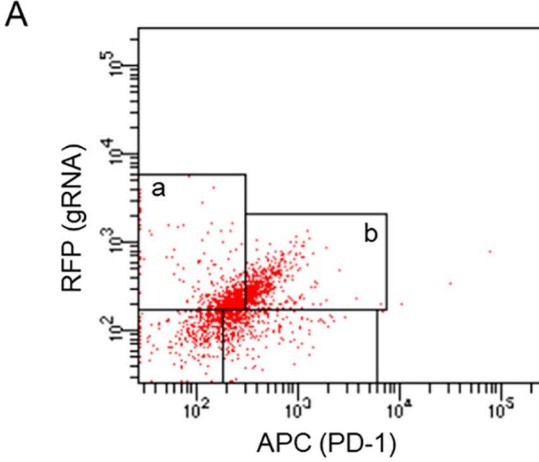

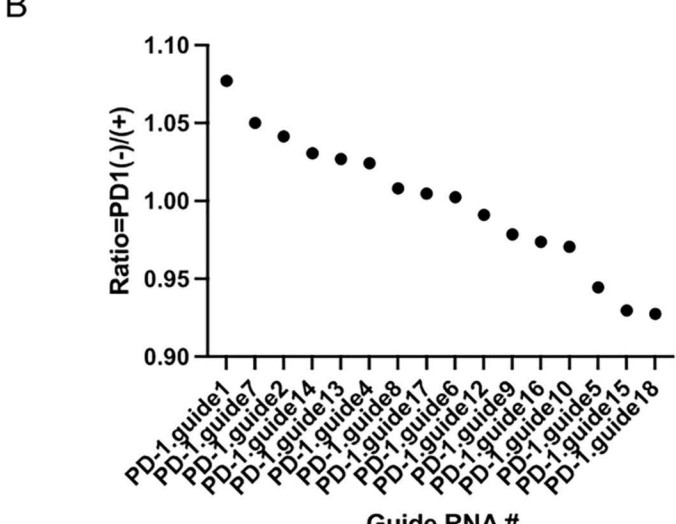

**Fig 2. Cell sorting of gRNA(+) cells. A)** Dot plots with gRNA and PD-1 parameters were used to define gRNA(+) cells in **(a)** PD-1-negative and **(b)** PD-1-positive populations. **b)** Next generation sequencing was then performed to determine the number of each gRNA. The number of reads for each gRNA was converted to number of reads per million mapped reads. The converted number of each gRNA region is shown as the ratio of gRNA-introduced PD-1-negative cells to PD-1-positive cells.

guide1 was highly abundant. These results indicated that guide1 targeted the Exon2 splice cis-element that regulates the splicing of PD-1 pre-mRNA.

**Effect of targeting the Exon2 splice cis-element in PD-1 on the production of interferon-γ (IFN-γ), TNF-α, IL-6, and CXCL10**

Porter and coworkers [11] reported delayed onset of cytokine secretion with vigorous *in vivo* chimeric antigen receptor T-cell expansion and prominent antileukemia activity, demonstrating substantial and sustained effector functions of CART19 cells. Therefore, in this study, we also examined the effect of targeting the Exon2 splice cis-element on the production of cytokines, which are indicators of anti-tumor activity. The amount of IFN-γ was significantly lower in RNA-targeted cells than in non-targeted cells (Fig 3A). The amount of IFN-γ in the culture supernatant of targeted cells was approximately 1230 pg/ml. Although less than in non-targeted cells, the result indicated that targeted CD8+ T cells retain the ability to produce IFN-γ following stimulation (Fig 3A). TNF-a is known to promote the differentiation of cytotoxic CD8+ T cells and eliminate tumors [12,13]. The amount of TNF-α was also significantly lower in RNA-targeted cells than in non-targeted cells (Fig 3B). The amount of TNF-α in the culture supernatant of targeted cells was approximately 570 pg/ml. Although less than in non-targeted cells, the result indicated that targeted CD8+ T cells retain the ability to produce TNF-α following stimulation (Fig 3B). IL-6 also promotes the differentiation of cytotoxic CD8+ T cells [14]. The amount of IL-6 was also significantly lower in RNA-targeted cells than in non-targeted cells (Fig 3C). The amount of IL-6 in the

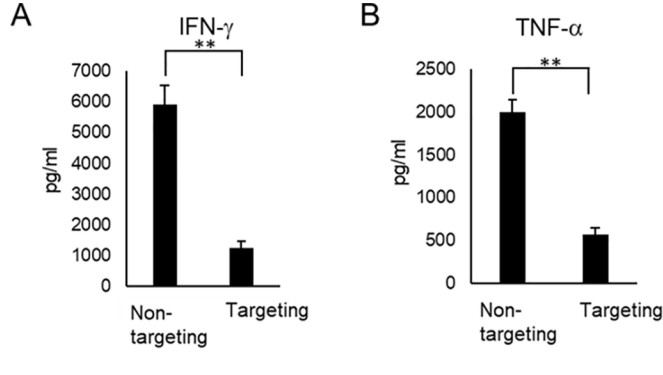

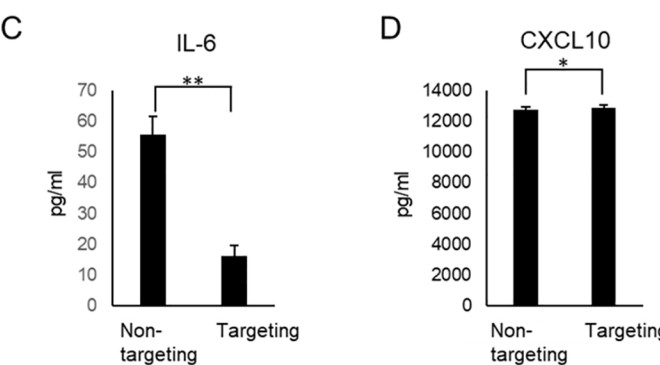

**Fig 3. Verification of the maintenance of cytokine and chemokine production ability related to cytotoxicity in CD8+ T cells targeting the cis-element.** Cytokine and chemokine levels were determined using a Luminex-based multiplex assay. The concentration (pg/ml) of IFN- g (A), TNF- a (B), IL-6 (C) and CXCL10 (D) is indicated in the case of control (non-targeting) and cells targeting the cis-element of Exon2 (targeting).

culture supernatant of targeted cells was approximately 20 pg/ml. Although less than in non-targeted cells, the result indicated that targeted CD8[+] T cells retain the ability to produce IL-6 following stimulation (Fig 3C). CXCL10 is a critical chemokine that attracts T cells into the tumor microenvironment [15]. Although the amount of CXCL10 was significantly higher in RNA-targeted cells than in non-targeted cells, the amount of CXCL10 in the culture supernatant of non-targeted and targeted cells was found to be comparable (Fig 3D). The result indicated that targeted CD8[+] T cells retain the ability to produce CXCL10 following stimulation (Fig 3D).

### Effect of targeting the Exon2 splice cis-element in PD-1 on the production of G-CSF, GM-CSF, IL-5, IL-8, and fractalkine

Granulocyte-colony stimulating factor (G-CSF) or granulocyte-macrophage colony-stimulating factor (GM-CSF), which promote neutrophil survival and activation, are known to induce adaptive antitumor immune responses and regression of established tumors based on neutrophil–T-cell interactions [16,17]. The amounts of G-CSF and GM-CSF were significantly lower in RNA-targeted cells than in non-targeted cells (Fig 4A, 4B). The amount of GM-CSF in the culture supernatant of targeted cells was approximately 3920 pg/ml. Although less than in non-targeted cells, the result indicated that targeted CD8[+] T cells retain the ability to produce GM-CSF following stimulation. It is known that IL-5 recruits eosinophils, which then activate CD8[+] T cells [18]. The IL-8–responsive CD8 T-cell subset was enriched in perforin, granzyme B, and IFN-γ, and had high cytotoxic potential [19]. There were no significant differences in the production of IL-5 and IL-8 among the RNA-targeted and non-targeted cells (Fig 4C, 4D). The amount of IL-5 and IL-8 in the culture supernatant of targeted cells was approximately 2480 pg/ml and 2740 pg/ml, respectively. The result indicated that targeted CD8[+] T cells retain the ability to produce IL-5 and IL-8 following stimulation. Fractalkine is a chemokine involved in the migration of cytotoxic T lymphocytes [20]. There were no significant differences in the production of fractalkine among the targeted and non-targeted cells (Fig 4E).

### Effect of targeting the Exon2 splice cis-element in PD-1 on the production of IL-4, IL-10, and IL-13

IL-4 is known to promote eosinophil expansion or migration [21]. The amount of IL-4 was significantly lower in RNA-targeted cells than in non-targeted cells (Fig 5A). The amount of IL-4 in the culture supernatant of targeted cells was approximately 1140 pg/ml. Although less than in non-targeted cells, the result indicated that targeted CD8[+] T cells retain the ability to produce IL-4 following stimulation. Classically, IL-10 is known to inhibit T cell responses. On the other hand, IL-10 enhances CD8[+] T cell proliferation, cytotoxic activity, and IFN-γ production [22]. IL-13 is also known to promote eosinophil expansion or migration. Eosinophils are expected to be of use as cellular biomarkers and effector cells in cancer therapy following ICI, especially with anti-CTLA4 and anti-PD-1 antibodies [21]. There were no significant differences in the production of IL-10 and IL-13 among the targeted and non-targeted cells (Fig 5B, 5C). The amount of IL-13 in the culture supernatant of targeted cells was approximately 10570 pg/ml (Fig 5C). The result indicated that targeted CD8[+] T cells retain the ability to produce IL-13 following stimulation.

## Discussion

The present study investigated guide1 RNA targeting of the Exon2 splice cis-element in pre-mRNA of PD-1 using 18 different guide RNAs. It is important that the guide RNA was extracted by screening in proliferating T cells using the CRISPR/dCas13 system. While targeting the Exon2 splice element in pre-mRNA of PD-1, cytokine secretion capacity was maintained in RNA-targeted CD8[+]T cells. For some cytokines, cytokine production levels were lower in RNA-targeted cells compared to non-targeted cells, which may imply that the RNA editing technology itself has some effect on the state of T cells.

The unique CRISPR/dCas13 technology used in this study allows for the biology-based search of drug targets on RNA. This makes it possible to take an approach that selectively inhibits only specific interactions with RNA involving various molecules without degrading the RNA [23]. In future, we would like to apply the RNA targeting method obtained in this

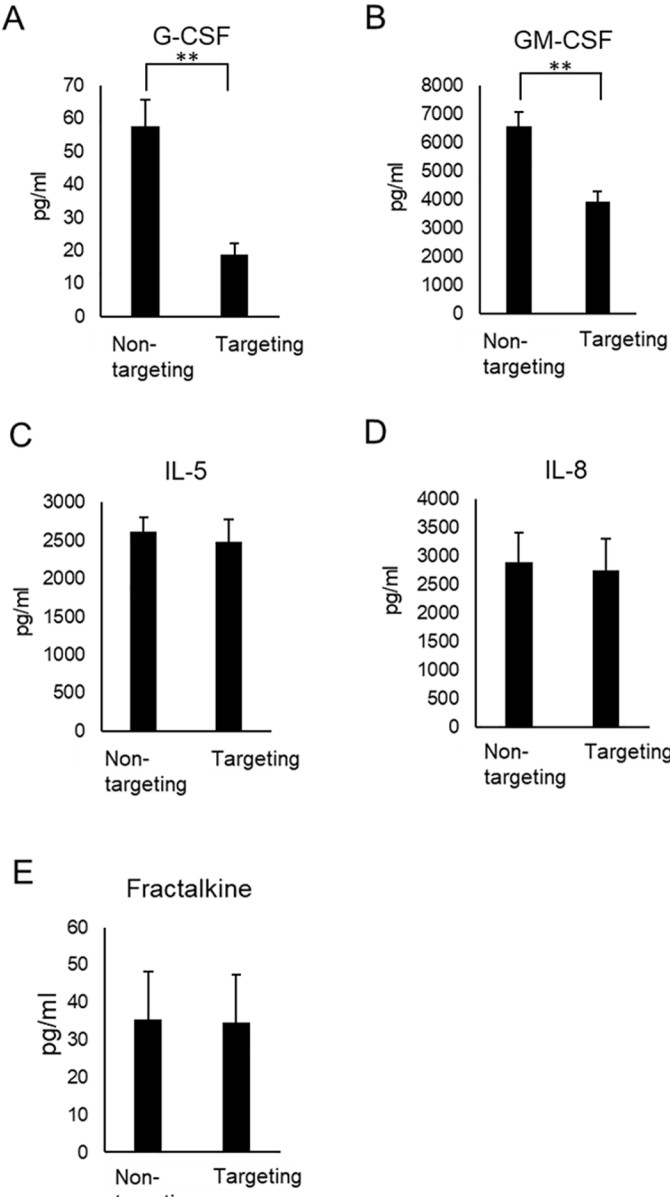

**Fig 4. Verification of the maintenance of cytokine and chemokine production ability involved in various immune responses in CD8⁺ T cells targeting the cis-element.** Cytokine and chemokine levels were determined using a Luminex-based multiplex assay. The concentration (pg/ml) of G-CSF **(A)**, GM-CSF **(B)**, IL-5 **(C)**, IL-8 **(D)** and fractalkine **(E)** is indicated in the case of control (non-targeting) and cells targeting the cis-element of Exon2 (targeting).

study to lymphocytes derived from ascites of cancer patients. By targeting PD-1 pre-mRNA, it will be possible to prevent lymphocytes from expressing the extracellular domain of PD-1. Ascites of cancer patients is often discarded with symptomatic treatment, but contains valuable lymphocytes from cancer patients. Therefore, we intend to extract these lymphocytes from ascites scheduled for disposal, target the pre-mRNA of lymphocytes, and then returning them to the cancer patient in an effort to contribute towards the promotion of anti-tumor activity. To do this, it is necessary to assess tumor

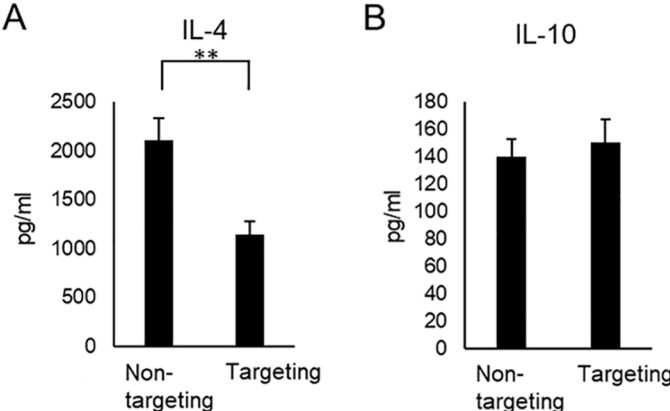

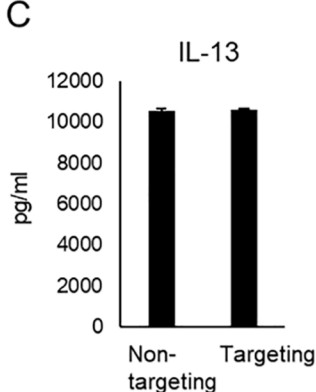

**Fig 5. Verification of the maintenance of cytokine production ability involved in various anti-immune responses in CD8⁺ T cells targeting the cis-element.** Cytokine levels were determined using a Luminex-based multiplex assay. The concentration (pg/ml) of IL-4 **(A)**, IL-10 **(B)** and IL-13 **(C)** is indicated in the case of control (non-targeting) and cells targeting the cis-element of Exon2 (targeting).

clearance capacity by conducting in vitro cell-killing assays or in vivo murine tumor model experiments, which has not yet been examined in present study.

Previous studies have been reported on exon 3 of PD-1 pre-mRNA in cancer cells [24], but our study differs in that we target Exon2 of PD-1 pre-mRNA in CD8⁺ T cells. PD-1 is known to be expressed in various cell types, so existing immunotherapy with anti-PD-1 antibodies may have unexpected effects depending on the cells. We used CD8⁺T cell lines in this study. If only the patient's lymphocytes are targeted, as in this study, PD-1 expression on lymphocytes could be suppressed, it might lead to enhanced antitumor activity. Further validation experiments using primary human T cells are needed.

## Supporting information

**S1 Table. A list of the guide RNA sequences.**
(PDF)

**S2 Table. Data the values used to build graphs.**
(PDF)

## Acknowledgments

We thank Dr. H. Asada for generously providing the EBT-8 cells.

## Author contributions

**Conceptualization:** Yuto Tan, Naoko Kumagai-Takei, Tatsuo Ito.

**Data curation:** Yuto Tan, Yurika Shimizu.

**Formal analysis:** Yuto Tan, Naoko Kumagai-Takei, Yurika Shimizu, Tatsuo Ito.

**Funding acquisition:** Naoko Kumagai-Takei.

**Investigation:** Yuto Tan, Naoko Kumagai-Takei.

**Methodology:** Yuto Tan, Naoko Kumagai-Takei, Yurika Shimizu, Mari Hara-Yamamoto, Tatsuo Ito.

**Project administration:** Naoko Kumagai-Takei.

**Resources:** Yurika Shimizu, Tatsuo Ito.

**Supervision:** Shigeru Mitani, Tatsuo Ito.

**Validation:** Mari Hara-Yamamoto.

**Visualization:** Akira Yamasaki.

**Writing – original draft:** Yuto Tan, Naoko Kumagai-Takei, Tatsuo Ito.

**Writing – review & editing:** Yuto Tan, Naoko Kumagai-Takei, Tatsuo Ito.

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
