## [Decision Letter · Decision Letter 0]

30 Jun 2025

Dear Dr. Kumagai-Takei,

We look forward to receiving your revised manuscript.

Kind regards,

Xianmin Zhu

Academic Editor

PLOS ONE

Journal Requirements:

“N.K.-T.

This work was supported by JPSS KAKENHI Grants (22K10497) and Kawasaki Medical School Project Grants (R03B-039, R04B-042, R05B-043, R06B-029).”

4. Please expand the acronym “N.K.-T. and JPSS KAKENHI” (as indicated in your financial disclosure) so that it states the name of your funders in full.

5. We note that your Data Availability Statement is currently as follows: All relevant data are within the manuscript.

Reviewers' comments:

Reviewer's Responses to Questions

**Comments to the Author**

1. Is the manuscript technically sound, and do the data support the conclusions?

Reviewer #1: Yes

2. Has the statistical analysis been performed appropriately and rigorously?

Reviewer #1: Yes

3. Have the authors made all data underlying the findings in their manuscript fully available?

Reviewer #1: Yes

4. Is the manuscript presented in an intelligible fashion and written in standard English?

Reviewer #1: Yes

Reviewer #1: This study addresses key limitations of current T-cell therapies—particularly Chimeric Antigen Receptor T-cell (CAR-T) therapy—including insufficient T-cell activation, poor in vivo persistence leading to incomplete tumor cell clearance and recurrence risks, and genotoxicity associated with conventional genome editing. It proposes an RNA editing strategy based on CRISPR/dCas13. Following PD-1 knockout, CD8⁺ T cells retain the ability to secrete critical cytokines (e.g., IFN-γ, TNF-α, GM-CSF) and proliferate.

However, significant problems exist in the article, as follows:

1. Invalidity of cross-system comparisons:

The methodology of directly contrasting cytokine concentrations in in vitro culture supernatants with serum data from other studies is unreasonable. Culture supernatants reflect transient local secretion, whereas serum is the result of systemic metabolic equilibrium. Culture supernatants cause cytokine accumulation due to lacking clearance mechanisms, while serum undergoes dynamic metabolic regulation. Absolute concentration comparisons are meaningless due to differences in kits, antibody affinity, and calibrator standards across studies.

2. Lack of evidence for functional inference:

Data indicate that PD-1-knockout cells retain in vitro secretory capacity but at levels far lower than non-target cells, demonstrating the knockout itself has substantially compromised T-cell status. Moreover, the authors did not assess tumor clearance capacity by conducting in vitro cell-killing assays or in vivo murine tumor model experiments. Therefore, cytokine secretion capacity cannot be equated with its actual effect within the in vivo tumor microenvironment (TME).

3. PD-1 mRNA knockout was conducted only in two T-cell lines in this study, not verified in primary T cells, resulting in lack of persuasiveness."

In summary, while the proposed CRISPR/dCas13 strategy offers a novel approach to mitigate genotoxicity, the study's conclusions regarding preserved T-cell function are undermined by flawed comparative analyses, insufficient functional validation, and the absence of primary T-cell data. Translationally relevant claims require rigorous validation in biologically appropriate models and assays.

**Do you want your identity to be public for this peer review?** For information about this choice, including consent withdrawal, please see our Privacy Policy

Reviewer #1: No

---

## [Author Response · Author response to Decision Letter 1]

9 Aug 2025

Response to Reviewer

We are grateful for the critical comments and useful suggestions, which helped us to improve our manuscript PONE-D-25-31015, entitled “Targeting the Exon2 splice cis-element in PD-1 and its effects on lymphocyte function.” As indicated in the following responses, we have taken all comments and suggestions into account and have improved the manuscript with five figures and one Supplementary Information.

Reviewer Comments to Author

Reviewer #1: This study addresses key limitations of current T-cell therapies—particularly Chimeric Antigen Receptor T-cell (CAR-T) therapy—including insufficient T-cell activation, poor in vivo persistence leading to incomplete tumor cell clearance and recurrence risks, and genotoxicity associated with conventional genome editing. It proposes an RNA editing strategy based on CRISPR/dCas13. Following PD-1 knockout, CD8⁺ T cells retain the ability to secrete critical cytokines (e.g., IFN-γ, TNF-α, GM-CSF) and proliferate.

However, significant problems exist in the article, as follows:

Comment 1:

1. Invalidity of cross-system comparisons:

The methodology of directly contrasting cytokine concentrations in in vitro culture supernatants with serum data from other studies is unreasonable. Culture supernatants reflect transient local secretion, whereas serum is the result of systemic metabolic equilibrium. Culture supernatants cause cytokine accumulation due to lacking clearance mechanisms, while serum undergoes dynamic metabolic regulation. Absolute concentration comparisons are meaningless due to differences in kits, antibody affinity, and calibrator standards across studies.

Response1:

In accordance with the reviewer’s comment, we agreed that comparisons with absolute cytokine levels from other studies, including serum measurement results from other studies, were not valid, so we performed statistical processing on the data obtained from these experiments (Fig. 3-Fig. 5) and interpreted the results.

In Abstract, we replaced the sentence “We extracted the RNA region of PD-1 pre-mRNA in the present study. By targeting this region of RNA, expression of the extracellular domain of PD-1 was specifically blocked while maintaining cytokine production and cell proliferation in CD8+ T cells.” with the sentence “We extracted the RNA region of PD-1 pre-mRNA using CD8+T cell lines and examined the effect of targeting the Exon2 splice cis-element on the production of cytokines in the present study. In particular, the production of IFN-�, TNF-�, GM-CSF was lower in RNA-targeted cells than in non-targeted cells, but the cytokine secretion capacity and cell proliferation were maintained in RNA-targeted cells.”.

See Abstract (p.3)

In Materials and Methods, we added about Statistical Analysis with the sentences “Significant differences were determined using Student's t-test and are indicated by asterisks (*P < 0.05, **P < 0.01).”

See Materials and Methods (p.9)

In the Results, about Figure 3, the following words and sentences were deleted:

We deleted the word “Figure 3A” from the sentence “Therefore, in this study, we also examined the effect of targeting the Exon2 splice cis-element on the production of cytokines, which are indicators of anti-tumor activity.”. We deleted the sentences “The amount of IFN-� exceeded the levels reported in a previous study (11).”, and “Although serum and bone marrow tumor necrosis factor α levels remained unchanged following CART19-cell infusion (11), the amount of TNF-� in the culture supernatant of targeted cells exceeded the levels reported with CART19-cell infusion (Figure 3B).”. We deleted the words “to previous reported levels in CART-cell fusion” from the sentence “The amount of CXCL10 in the culture supernatant of non-targeted and targeted cells was found to be comparable to previous reported levels in CART-cell fusion (Figure 3D).”

See Results (p.12-13)

In Results, about Figure 3, the following sentences were added:

We add the sentences “The amount of IFN-� was significantly lower in RNA-targeted cells than in non-targeted cells (Figure 3A).” and “The amount of TNF-� was also significantly lower in RNA-targeted cells than in non-targeted cells (Figure 3B). The amount of TNF-� in the culture supernatant of targeted cells was approximately 570 pg/ml. Although less than in non-targeted cells, the result indicated that targeted CD8+ T cells retain the ability to produce TNF-� following stimulation (Figure 3B).”, and “The amount of IL-6 was also significantly lower in RNA-targeted cells than in non-targeted cells (Figure 3C).” and “Although the amount of CXCL10 was significantly higher in RNA-targeted cells than in non-targeted cells, the”.

See Results (p.12-13)

In the Results, about Figure 4, the following words and sentences were deleted:

We deleted the sentences “The amount of G-CSF in the culture supernatant of non-targeted and targeted cells was found to be comparable to previous reported levels in the serum of healthy volunteers (Figure 4A) (18). The amount of GM-CSF in the culture supernatant of non-targeted and targeted cells was greater than in the serum of healthy volunteers (Figure 4B) (19).　The result indicated that targeted CD8+ T cells retain the ability to produce GM-CSF following stimulation.”. We deleted the sentence “The amount of IL-5 and IL-8 in culture supernatants of non-targeted and targeted cells was greater than in the serum of healthy volunteers (Figure 4CD), as previously reported (22).”. We deleted the sentence “The amount of fractalkine in the culture supernatant of non-targeted and targeted cells was found to be comparable to previous reported levels in the serum of healthy volunteers (Figure 4E) (24).”.

See Results (p. 14-15)

In Results, about Figure 4, the following sentences were added:

About Fig. 4, we add the sentences “The amounts of G-CSF and GM-CSF were significantly lower in RNA-targeted cells than in non-targeted cells (Figure 4AB). The amount of GM-CSF in the culture supernatant of targeted cells was approximately 3920 pg/ml.” We added the words “Although less than in non-targeted cells” before the sentences “the result indicated that targeted CD8+ T cells retain the ability to produce GM-CSF following stimulation.”. We added the sentences “There were no significant differences in the production of IL-5 and IL-8 among the RNA-targeted and non-targeted cells (Figure 4CD). The amount of IL-5 and IL-8 in the culture supernatant of targeted cells was approximately 2480 pg/ml and 2740 pg/ml, respectively.” And “There were no significant differences in the production of Fractalkine among the targeted and non-targeted cells (Figure 4E).”.

See Results (p.14-15)

In Results, about Figure 5, the following sentences were deleted:

About Fig. 5, we deleted the sentences “The amount of IL-4 in culture supernatants of non-targeted and targeted cells was greater than in the serum of healthy volunteers, as previously reported (26) (Figure5A).”, and “The amount of IL-10 in culture supernatants of non-targeted and targeted cells was found to be comparable to previous reported levels in the serum of healthy volunteers (Figure 5B) (28). The amount of IL-13 in culture supernatants of non-targeted and targeted cells was greater than in the serum of healthy volunteers, as previous reported (22) (Figure5C).”.

See Results (p.16)

In Results, about Figure 5, the following sentences were added:

About Fig. 5, we added the sentences “The amount of IL-4 was significantly lower in RNA-targeted cells than in non-targeted cells (Figure 5A). The amount of IL-4 in the culture supernatant of targeted cells was approximately 1140 pg/ml. Although less than in non-targeted cells, the result indicated that targeted CD8+ T cells retain the ability to produce IL-4 following stimulation.” and “There were no significant differences in the production of IL-10 and IL-13 among the targeted and non-targeted cells (Figure 5BC). The amount of IL-13 in the culture supernatant of targeted cells was approximately 10570 pg/ml (Figure 5C). The result indicated that targeted CD8+ T cells retain the ability to produce IL-13 following stimulation.”.

See Results (p.16)

In Discussion, we replaced the sentences “cytokine production was sufficient and T cell function was not regulated. This means that the guide1 RNA extracted using the CRISPR/dCas13 system specifically binds to the pre-mRNA region of PD-1 and does not affect other functions, such as cytokine production in immune cells.” with the sentences “cytokine secretion capacity was maintained in RNA-targeted CD8+T cells. For some cytokines, cytokine production levels were lower in RNA-targeted cells compared to non-targeted cells, which may imply that the RNA editing technology itself has some effect on the state of T cells.”.

See Discussion (p.18)

In References, we removed 6 references describing serum cytokine levels.

References (p.24-25)

In Figures, we have reflected the results of statistical processing in the figures 3-5.

See Figures 3-5

Comment 2:

2. Lack of evidence for functional inference:

Data indicate that PD-1-knockout cells retain in vitro secretory capacity but at levels far lower than non-target cells, demonstrating the knockout itself has substantially compromised T-cell status. Moreover, the authors did not assess tumor clearance capacity by conducting in vitro cell-killing assays or in vivo murine tumor model experiments. Therefore, cytokine secretion capacity cannot be equated with its actual effect within the in vivo tumor microenvironment (TME).

Response2:

In accordance with the reviewer’s comment, we agree that the knockout itself may be affecting the state of T cells. So, we added the sentences “For some cytokines, cytokine production levels were lower in RNA-targeted cells compared to non-targeted cells, which may imply that the RNA editing technology itself has some effect on the state of T cells.” in Discussion.

As pointed out, we have not evaluated the tumor elimination potential in in vitro cell killing assays or in vivo mouse tumor model experiments. Nevertheless, we apologize for the misleading statement that equates cytokine secretion potential with actual effects in the in vivo tumor microenvironment (TME). In Abstract, we replaced the sentence “In this study, we overcame the aforementioned problems by extracting the splice element of PD-1 pre-mRNA using biology based on CRISPR/dCas13.” with the sentence “By extracting the splice element of PD-1 pre-mRNA using biology based on CRISPR/dCas13 in this study, our ultimate goal is to overcome the above-mentioned challenges in the future.”. In Introduction, we replaced the sentence “In this study, we overcame the aforementioned problems by extracting the splice cis-element of PD-1 pre-mRNA using biology based on CRISPR/dCas13. Expression of the extracellular domain of PD-1 was specifically blocked while maintaining cytokine production and cell proliferation. It is expected that the use of RNA editing technology that targets only mRNA maturation should provide safe novel T cell therapy in the absence of genotoxicity.” with “We also examined the effect of targeting the Exon2 splice cis-element on lymphocyte function, focusing particularly on cytokine production.”. In Discussion, we added the sentences “To do this, it is necessary to assess tumor clearance capacity by conducting in vitro cell-killing assays or in vivo murine tumor model experiments, which has not yet been examined in present study.”.

See Abstract (p. 3), Introduction (p.5), Discussion (p.18-19).

Comment3:

3. PD-1 mRNA knockout was conducted only in two T-cell lines in this study, not verified in primary T cells, resulting in lack of persuasiveness."

Response3:

As pointed out, in this study, PD-1 mRNA knockout was performed only on two T cell lines. We are in the preparation stage to begin further research using primary T cells, but due to confidentiality concerns regarding our collaborative research, we only wish to include experiments using cell lines in this study. Therefore, in Abstract, we deleted the sentences “These results suggested that the use of the RNA editing technology maintains the safety of novel T cell therapy without genotoxicity by only targeting mRNA maturation.”. Instead, in Discussion, we added the sentences “we used CD8+T cell lines in this study. If only the patient's lymphocytes are targeted, as in this study, PD-1 expression on lymphocytes could be suppressed, it might lead to enhanced antitumor activity. Further validation experiments using primary human T cells are needed.”.

See Abstract (p. 3), Discussion (p. 19)

Comment4:

In summary, while the proposed CRISPR/dCas13 strategy offers a novel approach to mitigate genotoxicity, the study's conclusions regarding preserved T-cell function are undermined by flawed comparative analyses, insufficient functional validation, and the absence of primary T-cell data. Translationally relevant claims require rigorous validation in biologically appropriate models and assays.

Response4:

In accordance with the comments, we agree that rigorous validation in biologically relevant models and assays is required to make claims of translational significance. While we focused on cytokine secretion and cell proliferation, we understand that future claims of translational significance require experiments using those appropriate models, assays, and primary T cells. Therefore, we have revised our manuscript as shown in Responses 1-3 above.

---

## [Decision Letter · Decision Letter 1]

18 Aug 2025

Targeting the Exon2 splice cis-element in PD-1 and its effects on lymphocyte function.

PONE-D-25-31015R1

Dear Dr. Kumagai-Takei,

We’re pleased to inform you that your manuscript has been judged scientifically suitable for publication and will be formally accepted for publication once it meets all outstanding technical requirements.

Kind regards,

Xianmin Zhu

Academic Editor

PLOS ONE

Additional Editor Comments (optional):

Reviewers' comments:

Reviewer's Responses to Questions

**Comments to the Author**

Reviewer #1: All comments have been addressed

2. Is the manuscript technically sound, and do the data support the conclusions?

Reviewer #1: Yes

3. Has the statistical analysis been performed appropriately and rigorously?

Reviewer #1: Yes

4. Have the authors made all data underlying the findings in their manuscript fully available?

Reviewer #1: Yes

5. Is the manuscript presented in an intelligible fashion and written in standard English?

Reviewer #1: Yes

Reviewer #1: The authors have addressed all my concerns. Now this manuscript is ready to be published in the journal of PlosOne.

**Do you want your identity to be public for this peer review?** For information about this choice, including consent withdrawal, please see our Privacy Policy

Reviewer #1: **Yes: ** Haopeng Wang

---

## [Editor Report · Acceptance letter]

PONE-D-25-31015R1

PLOS ONE

Dear Dr. Kumagai-Takei,

I'm pleased to inform you that your manuscript has been deemed suitable for publication in PLOS ONE. Congratulations! Your manuscript is now being handed over to our production team.

Kind regards,

on behalf of

Dr. Xianmin Zhu

Academic Editor

PLOS ONE